# Identification and Characterization of Antiviral Activity of Synthetic Compounds Against Mayaro Virus

**DOI:** 10.3390/ph18050717

**Published:** 2025-05-13

**Authors:** Ana Paula Andreolla, Andrea Cristine Koishi, Alessandra Abel Borges, Larissa Albuquerque de Oliveira, Viviane Guedes de Oliveira, Nerilson Marques Lima, Eloah Pereira Ávila, Pedro Pôssa de Castro, Giovanni Wilson Amarante, Mauro Vieira de Almeida, Juliano Bordignon, Claudia Nunes Duarte dos Santos

**Affiliations:** 1Laboratório de Virologia Molecular, Instituto Carlos Chagas, ICC/Fiocruz, Rua Prof. Algacyr Munhoz Mader 3775, Cidade Industrial de Curitiba, Curitiba 81350-010, Paraná, Brazil; anaandreolla@hotmail.com (A.P.A.); andrea.koishi@fiocruz.br (A.C.K.); 2Laboratório de Pesquisas em Virologia e Imunologia, Instituto de Ciências Biológicas e da Saúde (ICBS), Universidade Federal de Alagoas (UFAL), Av. Lourival Melo Mota, s/n, Tabuleiro do Martins, Maceió 57072-900, Alagoas, Brazil; alessandra.borges@icbs.ufal.br; 3Departamento de Química, ICE, Universidade Federal de Juiz de Fora, Rua José Lourenço Kelmer s/n, Martelos, Juiz de Fora 36036-900, Minas Gerais, Brazil; larissalbuquerque24@gmail.com (L.A.d.O.); nerilsonmarques@gmail.com (N.M.L.); elo.avila@yahoo.com.br (E.P.Á.); pedro.possa@ufjf.br (P.P.d.C.); giovanni.amarante@ufjf.br (G.W.A.); mauro.almeida@ufjf.br (M.V.d.A.); 4Instituto de Educação, Agricultura e Ambiente, IEAA, Universidade Federal do Amazonas, Rua 29 de agosto, Centro, Humaitá 69800–000, Amazonas, Brazil; vivianeguedes.oliveira@gmail.com

**Keywords:** arbovirus, alphavirus, Mayaro virus, antiviral drugs, iHTS assay, flavonoids

## Abstract

**Background/objectives:** In Brazil, the co-circulation of arboviruses—such as dengue, Zika, yellow fever, and Chikungunya viruses—creates a complex epidemiological landscape, drawing attention from health authorities due to high morbidity and mortality rates. Also present in this context is the Mayaro virus (MAYV), a neglected arbovirus, which can also cause severe syndromes and has been expanding beyond its usual endemic areas in northern and central-western Brazil. Epidemiological surveillance measures remain limited, and there are no effective prophylactic strategies or antiviral treatments for this neglected arbovirus. In this study, we evaluated the antiviral activity of commercial and synthetic compounds against MAYV using an image high-throughput screening (iHTS) system. **Methods:** A total of 52 compounds from an FDA-approved commercial library (Tocriscreen) and 50 other compounds were tested. **Results:** Seven compounds showed anti-MAYV activity and were non-toxic for the following cell lines: Naringenin, LLA9A, chrysin, and its ester C6. Post-infection treatments with these selected compounds significantly decreased the percentage of infected cells and the release of infectious viral particles in the supernatant. Additionally, anti-MAYV activity of these four selected hits was confirmed using several human cell lines and two different MAYV genotypes. **Conclusions:** Our results indicate that the iHTS platform is effective for screening anti-MAYV drugs and that four promising compounds can efficiently inhibit MAYV replication in human cell lines. Although in vivo studies are still required to confirm the efficacy of the selected hits, our findings provide a starting point for developing a potential treatment for MAYV infections.

## 1. Introduction

Mayaro virus (*Alphavirus mayaro*, MAYV) is an arbovirus belonging to the family Togaviridae. Although it is considered a neglected disease, the number of human infections has gradually increased over the last few decades in different regions of the world, especially in the Americas [1,2,3,4,5] and particularly in Brazil [6,7,8,9,10,11]. MAYV is classified into three genotypes: genotype ‘D’, which is widely dispersed; genotype ‘L’, which has limited circulation and is primarily detected in Brazil; genotype ‘N’, detected only in Peru; and a recombinant D/L in Brazil and Haiti [12,13,14].

MAYV is primarily transmitted by *Haemagogus janthinomys* in a sylvatic cycle. However, the virus spreads primarily due to anthropogenic factors beyond forested and endemic rural areas. As a result, more spillover events to humans have been observed, increasing the potential for the virus to emerge and cause epidemics [10,15,16,17,18,19].

A large-scale epidemic appears plausible, as most of the population has not yet been exposed to MAYV. Furthermore, the experimental demonstration of vector competence in urban mosquitoes such as *Aedes aegypti* and the rural–urban bridge vector *Aedes albopictus* to transmit MAYV supports this possibility [20,21].

However, the close phylogenetic and antigenic relationship between MAYV and the highly prevalent Chikungunya virus (CHIKV) in several South American countries suggests that cross-protection against MAYV, mediated by the humoral response induced by prior CHIKV infection [22,23,24], could potentially influence the transmission dynamics and pathogenicity of MAYV.

In the acute phase, MAYV infection presents with symptoms similar to those of other arboviruses, including fever, chills, dizziness, myalgia, itching, and, most prominently, myalgia and arthralgia [8,25]. In some cases, the disease may progress to a chronic phase, with myalgia and arthralgia persisting for months or even years after the acute phase [8,25]. Other concerning complications, such as myocarditis and hemorrhagic or neurological manifestations may occur [25,26,27,28,29,30,31]. Due to its clinical similarity to other arboviruses, including Dengue virus (DENV) and CHIKV, MAYV infections are often underreported or misdiagnosed. Retrospective studies have indicated that MAYV infections occurred during DENV and other arbovirus outbreaks [1,18,19,32].

Molecular data indicate a clear prevalence and dispersion of MAYV across various regions in recent years, as well as its involvement in human infections in urban areas, highlighting its epidemic potential [19,33]. Despite the potential threat posed by MAYV in tropical and sub-tropical regions globally, no prophylactic measures or antiviral strategies for the prevention and/or treatment of MAYV infections are currently available [34,35]. The clinical approach is limited to symptomatic management, involving the administration of analgesics and antipyretics alongside supportive care based on the clinical presentation of infected patients.

Although in vitro studies have explored synthetic chemical substances, natural extracts, and commercial compounds as potential antiviral agents against MAYV [7,18,36,37,38,39,40], few candidates have progressed to preclinical and/or clinical trials [34,35]. In this context, we developed and validated a high-throughput imaging assay (iHTS) for in vitro screening of compounds with potential anti-MAYV activity and tested a library of compounds to evaluate their efficacy.

Over the last decades, carbamates have emerged as promising candidates in antiviral drug research due to their efficacy against a variety of viral pathogens [41]. Several commercially available antiviral drugs incorporate carbamate moieties in their structures, including efavirenz, ritonavir, amprenavir, atazanavir, and darunavir [42,43]. Notably, many of these carbamate-based antivirals also feature amino acid residues or analogs in their molecular frameworks, which are key to enhancing their bioactivity and specificity [42,43,44]. Drawing inspiration from this structural characteristic, we hypothesized that small carbamate-containing amino acid analogs could similarly exhibit antiviral properties.

Natural products, mainly secondary metabolites extracted from plants, remain a vital source for discovering and developing new antiviral drugs [45,46,47,48]. Among these, a class of phenolic compounds with diverse bioactivity is widely found in citrus fruits and has garnered significant attention [49,50,51]. Naringenin is a flavonoid of great biological importance, exhibiting activity against several diseases. Notably, its antiviral properties against DENV and Zika virus (ZIKV) have been well documented [45,50,51,52,53,54].

Chrysin, another flavonoid with reported antiviral activity, is known for its broad range of biological activities, including antiviral effects against influenza A and hepatitis B [55,56]. Its halogenated derivatives have been identified as potential inhibitors of DENV ZIKV infectivity, which have demonstrated high efficacy and low cytotoxicity and have therefore generated significant interest [11]. In this work, we describe the biological evaluation of 52 compounds from an FDA-approved commercial library (Tocriscreen) and 50 synthetic derivatives containing either a carbamate group or flavonoid moieties for their anti-MAYV activity.

## 2. Results

### 2.1. Standardization of Antiviral Screening Test for MAYV in iHTS

The cell lines C6/36, Vero E6, Huh-7.5, A172 and A549 were permissive to infection by MAYV_D. They exhibited productive infections, which resulted in a cytopathic effect, evidenced by a reduction in the number of nuclei in infected cell cultures compared to the MOCK (Appendix A). In contrast, the THP-1 cell line and differentiated THP-1 macrophages were refractory to infection (Appendix A).

The Huh7.5 cell line was chosen for the screening tests because, in addition to being permissive to MAYV, it is biologically relevant to the metabolism of the compounds. Various parameters were analyzed for the standardization of the iHTS assay for screening anti-MAYV compounds, such as cell density (1 and 2 × 10^4^ cells/well), MOIs (12.5, 2.5, 0.5, 0.1, and 0.02), and infection times (24 and 48 hpi; Appendix A). The infection conditions were optimized to identify the highest MOI and shortest incubation period, where no reduction in the number of nuclei was observed compared to MOCK (Appendix A). An MOI of 0.5 for 24 h was determined to be the optimal infection condition for Huh7.5 cells, with a mean and standard deviation of 25.31% ± 4.4 for 1 × 10^4^ cells/well and 33.7% ± 6.3 for 2 × 10^4^ cells/well. The mean infection value and Z index for both cell densities, using the statistical reliability test (Appendix A), were 29.87% infected cells and a Z score of 0.8304 for 1 × 10^4^ cells/well, and 44.84% infected cells and a Z score of 0.9044 for 2 × 10^4^ cells/well. Therefore, 2 1 × 0^4^ cells/well were selected for further assays.

Ribavirin and IFN-α 2a were tested at different concentrations to determine those inhibiting more than 80% of viral infection by MAYV_D while maintaining at least 80% CC_50_ based on nuclear quantification. The concentrations of 20 µM ribavirin and 1000 IU/mL IFN-α 2a were selected as positive controls for anti-MAYV activity in all subsequent in vitro tests (Appendix A).

Additionally, two protocols of viral infection were compared. In the first (method A), the viral inoculum was removed after the viral adsorption period to the cells. In the second (method B), the viral inoculum was maintained throughout the experiment, with subsequent administration of compounds and without washing out the viral inoculum. When using the two compounds (LLA10A and CSP1B) in both methodologies, method B showed a lower standard deviation (Appendix A). Furthermore, ribavirin and IFN-α 2a controls induced significant differences in antiviral activity between the two methodologies, based on an analysis of the percentage of infection (Appendix A). Method B maintained an 80% reduction in viral replication. Therefore, it was selected for anti-MAYV screening.

### 2.2. Standardization of MAYV Infection in Huh7.5 Cells by Flow Cytometry

Different parameters were considered for the standardization of MAYV_D infection using flow cytometry, including MOIs (0.02, 0.1, and 0.5) and infection times (24 and 48 hpi) (Appendix A). The optimal conditions were determined by comparing the percentages of infection with those obtained in the iHTS system. An MOI of 0.5 per 24 h was identified as the optimal infection condition for Huh7.5 cells in flow cytometry assays, resulting in a mean infection percentage of 45.8% ± 6.5.

### 2.3. Screening of Anti-MAYV Antiviral Compounds

A total of 104 compounds were tested for their anti-MAYV activity using iHTS. These compounds included flavonoids (flavanones, such as naringenin and its derivatives, n = 21; Table 1; flavones, such as chrysin and its derivatives, n = 9; Table 2), carbamates (n = 22; Table 3), and purified compounds from the Tocriscreen library (n = 52), which are already approved for clinical use in humans (Table 4). Of these, seven compounds at different concentrations beyond antiviral activity controls showed more than an 80% reduction in infection while maintaining over 80% CC_50_: naringenin, LLA5A, LLA5B, LLA9A, chrysin, ester C6, and PPC105 (Figure 1A). Regarding the Tocriscreen library, none of the tested compounds met the criteria required for selection as hits and were not further tested.

For the seven compounds initially selected in the screen, the concentration/response curve, as well as the CC_50_, IC_50_, and SI values, were determined (Figure 1B). The compounds with higher SI values were selected for further characterization of anti-MAYV activity: naringenin, LLA9A, chrysin, and ester C6 (Appendix A). Due to unsatisfactory SI values, LLA5A and LLA5B were discarded (Appendix A).

### 2.4. MNTC Determination

The MNTC of the compounds to the cells was determined using two assays: iHTS and apoptosis (7-AAD/Annexin V) through flow cytometry. In the first assay, six concentrations (400, 200, 100, 50, 25, and 12.5 µM) of the selected compounds (naringenin, LLA9A, chrysin, and ester C6) were tested in iHTS. The concentrations that showed more than 80% CC_50_, as determined by the nuclear quantification method (DAPI labeling), were considered nontoxic doses. From these results, the highest concentrations were selected as the MNTC. Accordingly, the MNTC for naringenin, LLA9A, chrysin, and ester C6 was 200, 50, 25, and 25 µM, respectively (Figure 2A).

To confirm the nontoxicity of these concentrations for Huh7.5 cells, we assessed the percentage of cell apoptosis (Annexin V) using flow cytometry. Huh7.5 cells were exposed to a range of concentrations for each compound and subsequently stained with Annexin V/7-AAD for flow cytometry analyses (Figure 2B). The results confirmed that the concentrations determined by the iHTS assay were indeed nontoxic for Huh7.5 cells and were therefore suitable for use in antiviral assays in this cell line (Figure 2B).

### 2.5. Virucidal Assay

The virucidal activity and interaction of the selected compounds with the virus particle were evaluated using virucidal assays, measured through viral titration and plaque formation at the MNTC for each compound. Viral titration revealed that the infection levels of C6/36 cells by MAYV, pre-incubated with the MNTC of the compounds, were comparable to those of the infection control. This result indicates that none of the six selected compounds exhibited virucidal activity against the MAYV particle (Appendix A).

### 2.6. Time Course Drug Administration Assay

The time course drug administration assay was performed to identify the stage of the viral replication cycle at which the compounds exhibit activity. Four conditions were tested based on the timing of drug administration: (1) pre-infection treatment of Huh7.5 cells (Figure 3A,B), (2) treatment during infection (Figure 3C,D), (3) treatment during/post infection (Figure 3C,D), and (4) treatment post infection (Figure 3C,D).

Results from the pretreatment times showed a progressive decrease in MAYV infection over time with treatment (Figure 3A,B). No significant differences were observed when the compounds were added simultaneously with the viral infection for any tested compound compared to the MAYV control (Figure 3C, red bar). However, significant differences were observed in post-infection treatments for all compounds compared to the infected control (MAYV; Figure 3C,D, purple bar). Furthermore, the percentage of viral infection was corroborated by virus titration results from the cell culture supernatants (Figure 3B,D).

### 2.7. Adsorption and Internalization Viral Assays

Adsorption and internalization assays were performed to explore the potential mechanisms of action of the selected compounds and confirm the lack of activity observed when the compounds were added during infection (Figure 3C,D). These assays aimed to determine whether the compound acts by blocking the binding of viral particles to host entry receptors (adsorption) or inhibiting the endocytosis of viral particles (internalization). The percentage of infection and viral particle production was evaluated in the assays to assess whether the compounds interfere with this stage of the MAYV viral cycle. The findings corroborate previous results, showing no significant differences in infection or viral particle production between the groups treated with the selected compounds and the infected control (Appendix A).

### 2.8. Antiviral Effectiveness of the Compounds in Established MAYV Infection in Cell Culture

The activity of the selected compounds against MAYV was evaluated at several hpi. For this purpose, Huh7.5 cells were infected and treated with compounds and ribavirin at 0, 2, 4, and 6 hpi (Figure 4A).

Naringenin reduced infection at all tested times (0, 2, 4, and 6 hpi) compared to the untreated infected control (Figure 4B). Similarly, LLA9A showed efficacy comparable to the ribavirin control up to 2 hpi; however, after this period, the efficacy of LLA9A decreased significantly (Figure 4B). Chrysin significantly reduced infection up to 4 hpi, while ester C6 exhibited activity only when administered immediately after infection, with no sustained effect at later administration times (Figure 4B).

Viral titration assays of supernatants from infected and treated cell cultures conducted during the same experiments revealed that the production of viable MAYV_D particles followed the same trends observed in IFI, yielding consistent results across both tests (Figure 4C). These findings, combined with the time course drug administration assays, suggest that the tested compounds likely act during early post-entry stages of the viral life cycle, such as replication, maturation, or viral particle assembly.

### 2.9. Antiviral Effectiveness Test for Different Cell Lineages

As MAYV infection can reach different organs in vertebrate hosts, we analyzed whether the antiviral effect of the selected anti-MAYV compounds is limited only to the initially tested cell line (Huh-7.5) or can act broadly in other cell lineages. In this context, we analyzed the anti-MAYV activity of the compounds during infection of human cell lines from lung (A549; Figure 5A,B) and neuronal (SH-SY5Y; Figure 5C,D) origins. All compounds (naringenin, LLA9A, chrysin, and ester C6) reduced the percentage of infected A549 and SH-SY5Y cells (Figure 5A–C) and viral particle production (Figure 5B–D) compared to the uninfected control. Thus, data indicates that the anti-MAYV activity of the compounds is cell line-independent.

### 2.10. Activity of Compounds Against Infection by Different MAYV Genotypes

Previous phylogenetic analysis revealed three MAYV genotypes: ‘D’, ‘L’, and ‘N’ [14]. To assess whether the effects of the selected compounds are genotype-specific, we first established infection parameters by analyzing the percentage of infected cells with two MAYV strains: genotype ‘D’ (MAYV_D; Figure 6B; blue bars) and genotype ‘L’ (BeAr20290; Figure 6B; red bars). Infection with the MAYV BeAr20290 strain in Huh7.5 cells was conducted using an MOI of 0.1 over 24 hpi.

The inhibitory activity of the anti-MAYV compounds was then assessed against this genotype. Except for LLA9A, which showed a significant effect only in MAYV_D infection, all other compounds (naringenin, chrysin, and ester C6) decreased the infection rate (Figure 6B) and viral particle production (Figure 6C) for both strains (Figure 6B). These findings were further corroborated through titration of the treated cell culture supernatants (Figure 6C). These results indicate that the anti-MAYV activity of naringenin, chrysin, and ester C6 is not genotype-specific.

## 3. Discussion

Until recently, MAYV sylvatic circulation and sporadic human outbreaks were primarily associated with the states of Pará and Amazonas in northern Brazil. However, enhanced epidemiological surveillance and the widespread availability of molecular diagnostic reagents have revealed the spread of MAYV to other regions of the country [2,14,18,38,57]. Currently, there are no immunoprophylactic strategies or antiviral treatments available to prevent MAYV infections [12]. In this context, investigating compounds with potential antiviral activity is critical for addressing this gap.

We found that some flavonoids, such as naringenin and chrysin, exhibit potential anti-MAYV activity in in vitro assays. The antiviral properties of these compounds have also been reported for other arboviruses, including CHIKV [58,59], DENV [51,60] and ZIKV [60,61]. Furthermore, flavonoids have shown antiviral effects against other human pathogens, such as the hepatitis C virus [62,63] and SARS-CoV-2 [64,65].

Additionally, modifications to the structure of these compounds significantly influenced both their cytotoxicity [45,60,66] and their activity against MAYV. Structural changes in naringenin and chrysin derivatives had variable effects on cell nuclear quantification, likely by inducing cell proliferation and toxicity, respectively [67].

Previous studies demonstrated that viral infection with Sinbdis virus and DENV was also affected, as treatment with some compounds decreased the infection rate while others increased the percentage of infected cells compared to the controls [68,69]. Structural alterations introduced during the synthesis of LLA9A, a naringenin derivative, reduced toxicity for both Huh7.5 and A549 cells. However, these modifications rendered the compound less effective in inhibiting MAYV and ZIKV infections [45,61] compared to naringenin.

The antiviral mechanisms of action of compounds can generally be categorized into three groups: (1) induction of antiviral responses within the cells, (2) interference with early stages of the viral cycle, such as adsorption, internalization, and uncoating, or (3) inhibition of viral replication, maturation, and release (including protein synthesis, viral genome replication, morphogenesis, and particle production and exit) [70]. A time point kinetics assay of drug administration was performed to determine which stage of the MAYV infection cycle was affected by the compounds selected in this study. This assay is widely used to evaluate the antiviral activity of compounds against various arboviruses, including MAYV [36,71], Venezuelan equine encephalitis [72], and DENV [51,66,73,74]. Pretreatment of Huh7.5 cells with the selected compounds (naringenin, chrysin, LLA9A, and chrysin ester C6) prior to infection with the MAYV_D genotype led to a reduction in infection over longer periods (24 h compared to two hours). A similar effect was observed when ZIKV-infected cells were pretreated with naringenin [61].

No effect was observed when Huh7.5 cells were treated with naringenin, chrysin, LLA9A, and chrysin ester C6 during infection, suggesting that the compounds do not interfere with the binding, adsorption, or internalization steps of MAYV in these cells. These findings align with those of previous studies on the antiviral effects of naringenin and chrysin against other relevant arboviruses, including CHIKV [58,59], Sindbis virus (SINV) [69], DENV [51,60] and ZIKV [60,61]. These studies indicate that the anti-DENV, anti-ZIKV, and anti-CHIKV activity of naringenin occurs after viral entry into cells [51,60,61], highlighting its ability to bind to the ZIKV protease (NS2b-NS3) [61] and CHIKV nsP3 [58,59]. Although the replication mechanism of MAYV has some unique features compared to CHIKV [34], the nsP4 proteins of MAYV and CHIKV—considered potential targets for drug therapy—exhibit substantial differences in shape, volume, and charge distribution. These structural variations suggest distinct interacting partners and biological functions, which could hinder the repurposing of CHIKV drugs for MAYV [71].

Except for LLA9A, all compounds tested in this study showed inhibitory activity against the two MAYV genotypes tested. Previous studies have shown effects of naringenin treatment on infection with the four DENV serotypes [51] and two ZIKV genotypes [61], with the outcome depending on the viral serotype.

The results of chrysin treatment, administered concomitantly with infection, for other arboviruses such as DENV, ZIKA [60], and CHIKV [59], support our findings on the inhibition of MAYV infection. We used two strains of MAYV: 1) MAYV_D, a clinical isolate, and 2) BeAr20290, a strain isolated in 1960 from *Hemagogus* spp. mosquitoes [75]. The difference in infection kinetics observed during treatment with LLA9A may be related to the passage history of the strains. The BeAr20290 strain, which has been cultured on various substrates since 1960, may have accumulated several genomic adaptive mutations, making it more resistant to treatments. In contrast, the MAYV_D strain, being a recent isolate and less adapted to cultivation, was more susceptible to the treatments. The NSP2 gene sequence of both genotypes differed in several amino acids, which could also explain the differences in susceptibility to treatments [76].

In this context, the BeAr20290 strain was less susceptible to LLA9A treatment, whereas it was more sensitive to treatments with other compounds. Considering that these two genotypes are the most prevalent in cases worldwide, advances in research on anti-MAYV compounds (naringenin and chrysin) for both are highly relevant. Thus, the hypothesis that these compounds act on a similar site is reinforced, suggesting that the MAYV viral protease, which is conserved between strains, could indeed be the target of these compounds [71].

The use of naringenin to treat MAYV infections is a promising option due to its anti-MAYV potential, as well as its analgesic and anti-inflammatory activities [52,77,78,79], which may help alleviate the arthralgia observed in patients infected with MAYV [32].

In this study, we demonstrated the effectiveness of four compounds (naringenin, LLA9A, chrysin, and C6 ester) in reducing MAYV infection rates and the production of infectious viral particles in vitro. The introduction of lipophilic groups at positions 7 and/or 4′ of naringenin or position 7 of chrysin to obtain alkylated and acylated compounds was aimed at increasing their lipophilic character, thereby enhancing interaction with cell membranes and improving permeability, which may lead to a better biological response [80,81,82]. Although treatment with chrysin was less efficient compared to the antiviral control (ribavirin), it showed promise in terms of antiviral activity compared to other substances, like naringenin. Chrysin is recognized for its hepatoprotective, antiallergic, anti-inflammatory, and antioxidant properties, which are relevant for controlling the symptoms associated with MAYV infections and other arthritogenic viruses [83,84,85]. Furthermore, these compounds are already on the market as dietary supplements, and their pharmacological safety has been established [62,86].

## 4. Methods

### 4.1. Cell Line and Viruses

C6/36 *Aedes albopictus* cells (ATCC CRL-1660) were cultured in Leibovitz’s L15 medium (Gibco, Grand Island, NE, USA) supplemented with 5% fetal bovine serum (FBS-Gibco), 25 µg/mL gentamicin (Gibco), and 0.27% tryptose at 28 °C. Vero E6 cells (Sigma, 85020206), A549 cells (ATCC CCL-185), Huh7.5 cells (ATCC PTA-8561), A172 cells (ATCC CRL-1620), and SH-SY5Y cells (ATCC CRL-2266) were maintained in Dulbecco’s Modified Eagle Medium/Nutrient Ham F12 (DMEM F12–Gibco) with 10% FBS, 14 mM sodium bicarbonate, 100 IU/mL penicillin (Sigma-Aldrich, Steinheim, Germany), and 100 µg/mL streptomycin (Sigma-Aldrich, Steinheim, Germany) at 37 °C in a 5% CO_2_ atmosphere. THP-1 cells (ATCC TIB-202) were cultured in Roswell Park Memorial Institute medium 1640 (RPMI-1640; Lonza, Basel, Switzerland) with 2 mM L-glutamine and supplemented with 10% FBS, 14 mM sodium bicarbonate, 100 IU/mL penicillin (Sigma-Aldrich,), and 100 µg/mL streptomycin (Sigma-Aldrich) at 37 °C in a 5% CO_2_ atmosphere.

MAYV_D is a clinical isolate of the ‘D’ genotype, kindly provided by Dr. Ana Maria Bispo de Filippis from the Virology Department, Oswaldo Cruz Foundation (Fiocruz), Rio de Janeiro, RJ, Brazil. MAYV BeAr20290 (GenBank accession number KT754168), belonging to the ‘L’ genotype, was generously supplied by Dr. Alessandra Abel Borges from the Virology and Immunology Research Laboratory (LAPEVI), Federal University of Alagoas, Maceió, AL, Brazil. This strain was transferred to the Molecular Virology Laboratory, Oswaldo Cruz Foundation (Fiocruz), Curitiba, PR, Brazil, where it was amplified and titrated using the plaque-forming assay in C6/36 cells [87,88].

### 4.2. Library of Compounds

Naringenin (purity > 95%; SC-203443; Santa Cruz Biotechnology, Dallas, TX, USA), chrysin (purity = 97%; C80105; Sigma-Aldrich, St. Louis, MO, USA), and ribavirin (Sigma-Aldrich) were diluted to 20 mM in 100% dimethyl sulfoxide (DMSO; Sigma-Aldrich). Interferon α-2A (IFN-α 2A; Blau Farmacêutica, Cotia, São Paulo, Brazil) was diluted to 1000 IU/mL. The other compounds used in the screening, including flavonoids derived from naringenin or chrysin and carbamate-containing compounds, were synthesized at the Federal University of Juiz de Fora, Juiz de Fora, MG, Brazil, and diluted in 100% DMSO (Table 1, Table 2 and Table 3). Naringenin and chrysin derivatives (Figure 1 and Figure 2) were prepared following procedures previously described in the literature [45,89,90]. The characterization data, for ether and the ester derivatives, are available in the literature [45]. The chalcone was obtained as a ring-opening isomerization reaction product, as described by Ávila et al. 2021 [89]. The chrysin data are described by Zhu et al. 2014 [90], Cheng et al. 2014 [91], and in SI.A series of compounds incorporating the carbamate group alongside amino acid scaffolds were synthesized according to literature protocols (Figure 3) [92,93,94]. The FDA-approved Tocriscreen library (compound number 5932) was acquired and stored at the Molecular Virology Laboratory, Oswaldo Cruz Foundation (Fiocruz), Curitiba, PR, Brazil, containing 159 compounds at 10 mM and diluted in 100% DMSO, as recommended by the manufacturer (Table 4).

### 4.3. Indirect Immunofluorescence Assay (IFA) for iHTS

For the IFA, 2 × 10^4^ Huh7.5 cells were seeded in a 96-well plate and incubated overnight. The cells were then infected with MAYV_D at a multiplicity of infection (MOI) of 0.5 in the presence or absence of the test compounds. Twenty-four hours post infection (hpi), cells were fixed with 200 µL/well of a methanol–acetone solution (1:1) and incubated at −20 °C for 1 h. One hundred microliters/well of the monoclonal antibody (mAb) 1G1, developed in-house against the E2 protein of CHIKV (which cross-reacts with MAYV), was diluted 1:100 in phosphate-buffered saline solution with 1% bovine serum albumin (1% PBS-BSA) and incubated for 1 h at 37 °C. After three washes with phosphate-buffered saline solution plus 0.05% Tween 20 (PBS-T), the anti-mouse antibody conjugated with Alexa Fluor 488 (1:1000) (Sigma-Aldrich) and 0.3 mM 4′,6-diamidino-2-phenylindole (DAPI) were added, and the cells were incubated for 1 h at 37 °C. The cells were then washed three more times, and 100 µL/well of PBS was added. Images were captured using the Operetta CLS High-Content Analysis System (PerkinElmer, Waltham, MA, USA) with an 8-field scan at 20× magnification using a non-confocal objective. The data were analyzed using Harmony 4.8 High Content Imaging and Analysis Software (https://www.revvity.com/br-en/product/harmony-5-2-office-revvity-hh17000019, accessed on 1 April 2025). Nuclear quantification and the percentage of MAYV_D infection were evaluated by staining with DAPI and Alexa Fluor 488, respectively. Normalized values were obtained as previously described [73].

### 4.4. Screening for the Anti-MAYV Activity of the Compounds

Following the standardization of the iHTS methodology, flavonoids, carbamates, and compounds from the FDA-approved library were screened at different concentrations (200, 100, and 50 µM) for their activity against MAYV_D infection. The MAYV_D strain was selected for the initial tests because it is a recent clinical isolate.

Ribavirin (20 µM) and IFN-α 2A (1000 IU/mL) were antiviral activity controls. Both the test compounds and the controls were added individually to the MAYV_D 0.5 MOI inoculum. The resulting mixtures (compound + virus) were then incubated with 2 × 10^4^ cells/well of Huh7.5, seeded in 96-well plates. At 24 hpi, the plates were fixed, and the IFA was performed. Images were captured, and infection and nuclear quantification were analyzed as previously described.

The first criterion for selecting compounds with anti-MAYV activity was based on a scatterplot of normalized results. Compounds that showed more than an 80% reduction in the percentage of viral infection and 80% cell viability (CC_50_), as determined by nuclei quantification, were selected. The selectivity index (SI; SI = CC_50_/IC_50_) for these compounds was calculated after dose–response (IC_50_) and CC_50_ assays. The IC_50_ and CC_50_ assays were performed using eight concentrations of each selected compound (800, 400, 200, 100, 50, 25, 12.5, and 6.25 µM) in the iHTS assay. From the SI values, the most effective compounds with anti-MAYV activity were identified, based on the highest SI values.

### 4.5. Determination of the Maximum Non-Toxic Concentration (MNTC) of Anti-MAYV Selected Compounds

To confirm the MNTC, a cell death assay was performed using Annexin-V (FITC) and 7-AAD (BD Biosciences, San Jose, CA, USA) markers. Huh7.5 cells (1 × 10^5^ cells per well in 24-well plates) were treated for 24 h with six concentrations of each selected compound (naringenin, LLA9A, chrysin, and ester C6). The concentrations tested varied based on results obtained from the DMNT assay via the iHTS system. For naringenin, the tested concentrations were 3200, 1600, 800, 400, 200, and 100 µM. For ALL9A, the concentrations were 800, 400, 200, 100, 50, and 25 µM. For chrysin and chrysin ester C6, the concentrations were 400, 200, 100, 50, 25, and 12.5 µM.

Temperature-induced damage was used as a cell death parameter by exposing cells to 56 °C for 30 min. Following incubation, cells were trypsinized and labeled with Annexin-V (FITC) and 7-AAD according to the manufacturer’s instructions. Data were collected using flow cytometry (FACS Canto II; BD Biosciences) and analyzed with FlowJo software, version 10 (FlowJo LLC, San Fransisco, CA, USA). Viable cells were identified as negative for both markers in the analysis.

### 4.6. Time Course Drug Administration Assay

Compounds with the highest SI values (naringenin, LLA9A, chrysin, and ester C6) were evaluated to determine their activity at different stages of the MAYV life cycle. A time course drug administration assay was conducted to identify the viral cycle phase where the anti-MAYV activity exhibits viral entry, replication, exit (compound added during and after infection), or modulation of the host cell response to infection.

To assess whether the compounds modulate the host cell response, cells were pretreated with the previously established MNTC of each compound for 2, 4, 8, 16, and 24 h before infection with MAYV. The MNTC of each compound was also applied during infection to evaluate interference with viral attachment, particle entry, or simultaneous stages. Additionally, compounds were added concurrently with the virus inoculum (during/post) or immediately after the virus inoculum was washed out (post infection). For all experiments, cells were stained at 24 hpi to evaluate their viability and MAYV infection using the iHTS assay, as previously described.

### 4.7. Antiviral Effectiveness Test in Established Infection

To assess whether the selected compounds effectively controlled ongoing infections, 2 × 10^4^ Huh7.5 cells were infected with MAYV_D at an MOI of 0.5 for 1 h and 30 min. After incubation, the inoculum was removed, and the cells were washed with PBS. The selected compounds at their respective MNTCs were added at different hpi (0, 2, 4, and 6 hpi). At 24 hpi, the cell culture supernatant was collected and stored at −80 °C for subsequent viral titration. The cells were then washed and stained to quantify CC_50_ and MAYV infection using the previously described iHTS assay.

### 4.8. Anti-MAYV Activity in Different Human Cell Lines

To determine whether the anti-MAYV selected compounds retain activity during the infection and replication phases of MAYV despite the cell type, we performed the iHTS assay using two human cell lines: A549 (human adenocarcinoma cells of the basal alveolar epithelium; ATCC CCL-185) and SH-SY5Y (human neuroblastoma cells; ATCC CRL-2266). The optimal infection condition for both cell lines was established as an MOI of 1 over 24 h.

A total of 2x10^4^ A549 and SH-SY5Y cells were seeded per well in 96-well plates and subsequently infected with MAYV_D. Following an incubation period of 1 h and 30 min, the cells were washed with PBS, and a culture medium containing each selected anti-MAYV compound at its MNTC was added. At 24 hpi, the cells were washed and fixed with methanol–acetone. The iHTS assay was then performed to assess CC_50_ and quantify MAYV infection. The cell culture supernatant was also collected for virus titration, as previously described.

### 4.9. Anti-MAYV Activity Against Different Viral Strains

To evaluate whether the anti-MAYV activity of the selected compounds is maintained against different MAYV strains, we tested their effectiveness in Huh7.5 cells infected with two MAYV genotypes. Specifically, the BeAr 20290 strain was used to optimize infection conditions. Various MOIs (12.5, 2.5, 0.5, 0.1, and 0.002) and incubation periods (24 hpi and 48 hpi) were assessed (Figure 6A). Based on these results, an MOI of 0.1 over 24 h was determined to be the optimal condition for infection with this MAYV genotype.

Huh7.5 cells were infected side-by-side with MAYV_D (MOI of 0.5) and BeAr20290 (MOI of 0.1) and treated with the selected compounds at their MNTC in the post-infection condition. At 24 hpi, the cells were washed and fixed with methanol–acetone, and the iHTS assay was conducted to assess CC_50_ and quantify MAYV infection. The cell culture supernatant was also collected and subjected to virus titration as previously described.

### 4.10. Statistical Analysis

The results were analyzed using Prism software (GraphPad 7 Software, San Diego, CA, USA). Statistical evaluation was performed utilizing analysis of variance (ANOVA) followed by Tukey’s test, with a significance level set at *p* < 0.05. Nonlinear regression analysis, followed by the variable slope model, was employed to calculate CC_50_ and IC_50_ values.

## Data Availability

The original contributions presented in this study are included in the article/Appendix A. Further inquiries can be directed to the corresponding authors.

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
