# Peer review of "Identification and Characterization of Antiviral Activity of Synthetic Compounds Against Mayaro Virus"

_pharmaceuticals, 2025, doi:10.3390/ph18050717_

Round 1
Reviewer 1 Report
Comments and Suggestions for Authors
Ana Paula Andreolla; Claudia Nunes Duarte dos Santos and co-workers have presented their study entitled “Identification and Characterization of Antiviral Activity of Synthetic Compounds Against Mayaro Virus.” In this article, the authors have evaluated the antiviral activity of both commercial and synthetic compounds against Mayaro virus (MAYV) using image-based high-throughput screening (iHTS). A total of 52 compounds from an FDA-approved commercial library (Tocriscreen) and 50 additional synthetic compounds were tested. The seven compounds demonstrated anti-MAYV activity and exhibited non-toxicity. These promising compounds were able to efficiently inhibit MAYV replication in human cell lines. However, in vivo studies are still required to confirm the efficacy of these selected hits. Overall, this approach appears scientifically sound and provides a solid starting point for the development of potential treatments for MAYV. Therefore, I believe this article is suitable for publication in a high-impact journal, after addressing the following point.
- In the total number of compounds screened against the MAYV virus using iHTS, including flavonoids (flavanones such as naringenin and its derivatives, n=21; flavones such as chrysin and its derivatives, n=9) and carbamates (n=22), the authors should provide data their characterization data (NMR and Mass spectra).
- In table 4. Tocriscreen FDA-approved compounds list, all the compound concentration is 10 mM solution in DMSO, it could be removed from the table and adjusted to the foot note.
- In typos, page 25, scheme 2, X, Br or Cl.
- In Schemes 1, 2, and 3, the authors should use R–X and R–COOH as the coupling partners, since all products are represented with R, R¹, and R², not with CH₃(CH₂)ₙX or CH₃(CH₂)ₙ
- Please carefully review the manuscript for typographical and formatting errors to improve the overall quality of presentation.
Author Response
Reviewer 1
Ana Paula Andreolla; Claudia Nunes Duarte dos Santos and co-workers have presented their study entitled “Identification and Characterization of Antiviral Activity of Synthetic Compounds Against Mayaro Virus.” In this article, the authors have evaluated the antiviral activity of both commercial and synthetic compounds against Mayaro virus (MAYV) using image-based high-throughput screening (iHTS). A total of 52 compounds from an FDA-approved commercial library (Tocriscreen) and 50 additional synthetic compounds were tested. The seven compounds demonstrated anti-MAYV activity and exhibited non-toxicity. These promising compounds were able to efficiently inhibit MAYV replication in human cell lines. However, in vivo studies are still required to confirm the efficacy of these selected hits. Overall, this approach appears scientifically sound and provides a solid starting point for the development of potential treatments for MAYV. Therefore, I believe this article is suitable for publication in a high-impact journal, after addressing the following point.
- In the total number of compounds screened against the MAYV virus using iHTS, including flavonoids (flavanones such as naringenin and its derivatives, n=21; flavones such as chrysin and its derivatives, n=9) and carbamates (n=22), the authors should provide data their characterization data (NMR and Mass spectra).
- We have included a sentence (Page 23, lines 461-465) to clarify that flavanone and flavone derivatives were reported by our research group and the respective characterization data are described in the literature. On the other hand, we added the characterization data for compounds not yet reported.
- In table 4. Tocriscreen FDA-approved compounds list, all the compound concentration is 10 mM solution in DMSO, it could be removed from the table and adjusted to the foot note.
- We added concentration as a footnote.
- In typos, page 25, scheme 2, X, Br or Cl.
- It is corrected now in both Schemes 1 and 2.
- In Schemes 1, 2, and 3, the authors should use R–X and R–COOH as the coupling partners, since all products are represented with R, R¹, and R², not with CH₃(CH₂)ₙX or CH₃(CH₂)ₙ
- It was correct in both Schemes 1 and 2
- Please carefully review the manuscript for typographical and formatting errors to improve the overall quality of presentation.
- Thank you to point it out. We will check it throughout the manuscript.
Reviewer 2 Report
Comments and Suggestions for Authors
I have reviewed the manuscript and suggest some major revision
- Does the abstract clearly summarize the purpose, methods, key findings, and significance of the study?
- the main antiviral compounds studied explicitly named should be added in the abstract?
- Does the introduction provide adequate background on Mayaro virus and its importance? Please add rational study of the current work
- the public health relevance of MAYV infection and its similarities to other arboviruses well should be explained? Does the introduction justify the need for screening new compounds?
- The previous studies on antiviral agents against MAYV sufficiently reviewed and compare it with your current work.
- The novelty of your study should be highlighted in the introductionMethodology
- The cell lines used appropriate for this type of study. The iHTS assay protocol should be clearly explained and reproducible?
- The appropriate control compounds (e.g., ribavirin, IFN-α) be included in the manuscript.
- The procedure for compound dilution and treatment consistent across tests? Are the flow cytometry and apoptosis assays described in sufficient detail?
- The method for calculating CC50, IC50, and SI values should be adequately explained in the MS.
- The statistical analysis should be properly explained appropriately for the type of data collected.
- The virucidal, adsorption, and internalization assays should clearly be defined and justified?
- The procedures for testing multiple MAYV genotypes sound should be discuss properly.
- the findings on antiviral activity statistically are significant and well-supported. Do the selected compounds show consistent effects across different tests? Please comaptre your results with reported one
- the absence of virucidal activity should be clearly interpreted and explained
- Does the differences in compound effectiveness across time-points clearly visualized?
- Does the results across different human cell lines support the main conclusions?
- Does the discussion adequately interpret the results in light of the hypothesis?
- Are the antiviral mechanisms of action discussed in a meaningful way? Is the comparison with previous studies thorough and accurate?
- Are the roles of flavonoids and carbamates discussed in terms of structure-activity relationship? Are alternative explanations for the data considered?
- At page-25, there are three -OH group, How you can protect the remaining two -OH group of naringenin and chrysin derivatives.
- Did you characterize the synthesized compound through HNMR, CNMR, HRMS, and FTIR. Please provide their spectral data of all synthesized compounds.
Author Response
Reviewer 2
I have reviewed the manuscript and suggest some major revision
- Does the abstract clearly summarize the purpose, methods, key findings, and significance of the study? The main antiviral compounds studied explicitly named should be added in the abstract?
- It has been added to the abstract.
- Does the introduction provide adequate background on Mayaro virus and its importance?
Please add rational study of the current work the public health relevance of MAYV infection and its similarities to other arboviruses well should be explained.
Does the introduction justify the need for screening new compounds?
- Thank you for the considerations. We stressed these points in the introduction and discussion sections:
Mayaro virus (MAYV), the causative agent of Mayaro fever, is a neglected virus widely dispersed in Brazil and South America circulating mainly in enzootic cycles among sylvatic mosquitoes and nonhuman primates and other mammals. Nevertheless, some experimental studies suggest that MAYV could establish an urban cycle involving humans and Aedes aegypti and Ae. albopictus mosquitoes (https://doi.org/10.4269/ajtmh.22-0777).
While MAYV outbreaks are typically confined to northern and central-western states in Brazil, in recent years the virus is being detected in other regions of the country, following the same spread pathway observed recently for Oropouche virus epidemics in the Americas and Caribe.
As for most neglected tropical diseases, there are no immunoprophylactic strategies or antiviral treatments available to prevent MAYV infections. Aiming to feel this gap, in this study, we evaluated the antiviral activity of commercial and synthetic compounds against MAYV using an image high-throughput screening (iHTS) system and identified four promising compounds which can efficiently inhibit MAYV replication in human cell lines.
- The previous studies on antiviral agents against MAYV sufficiently reviewed and compare it with your current work.
- Besides the published articles cited in the manuscript, we included our recent published review “Mayaro Virus: The State-of-the-Art for Antiviral Drug Development” which compilates all articles related to antivirals and Mayaro virus [1]. However, none of them focused on flavonoids, although these compounds have been shown to be effective against other viruses.
- Andreolla, A.P.; Borges, A.A.; Bordignon, J.; Duarte dos Santos, C.N. Mayaro Virus: The State-of-the-Art for Antiviral Drug Development. Viruses 2022, 14, 1–21, doi:10.3390/v14081787.
- The novelty of your study should be highlighted in the introduction Methodology
- Thank you for the suggestion. We included a statement.
- The cell lines used appropriate for this type of study. The iHTS assay protocol should be clearly explained and reproducible?
- The rational for the cell lines choice was to reproduce as close as possible the natural target cells during MAYV infection and select the more informative in vitro systems for the drugs screening.
We used Huh7.5, which is a human and hepatic derived cell lineage. The choice of this lineage was based on: 1) hepatic metabolism of the compound, 2) the high rate of drug withdrawal due to hepatotoxicity, 3) it is a known site of viral replication. We also used other human derived lineages such as A549 (epithelium) and SH (neuronal).
Related arboviruses, such as Chikungunya virus, have been implicated in cases of fulminant hepatitis (https://pmc.ncbi.nlm.nih.gov/articles/PMC5330729/) and Guillan Barré syndrome cases.
For MAYV, the liver is also a site of virus replication “The virus replicates in white blood cells (e.g., monocytes, macrophages) and spreads to bones, muscles and joints via the main sites of replication, the spleen and the liver". Hence, the choice of this specific strain. At the same time, we were able to evaluate adverse toxicity situations in addition to efficiency against the virus in a well-known infection site. https://doi.org/10.3390/pathogens9090738.
The iHTS assay is a widely screening method used in pharmaceutical industry. It is well established because, in addition to analyzing the infection itself, is able to concomitantly evaluate other aspects, such as cellular integrity, i.e., pre-apoptotic events can be seen with nuclear staining intensity and nuclear size (both aspects demonstrate nuclear condensation). It is also able to differentiate between cell multiplication and cell death events, though the counting of nuclei numbers [2–4].
- 2. Qing, M.; Liu, W.; Yuan, Z.; Gu, F.; Shi, P.Y. A High-Throughput Assay Using Dengue-1 Virus-like Particles for Drug Discovery. Antiviral Res. 2010, 86, 163–171, doi:10.1016/j.antiviral.2010.02.313.
- Martin, H.L.; Adams, M.; Higgins, J.; Bond, J.; Morrison, E.E.; Bell, S.M.; Warriner, S.; Nelson, A.; Tomlinson, D.C. High-Content, High-Throughput Screening for the Identification of Cytotoxic Compounds Based on Cell Morphology and Cell Proliferation Markers. PLoS One 2014, 9.
- Abraham, V.C.; Towne, D.L.; Waring, J.F.; Warrior, U.; Burns, D.J. Application of a High-Content Multiparameter Cytotoxicity Assay to Prioritize Compounds Based on Toxicity Potential in Humans. J. Biomol. Screen. 2008, 13, 527–537, doi:10.1177/1087057108318428.
- The appropriate control compounds (e.g., ribavirin, IFN-α) be included in the manuscript.
- We thank your suggestion. These compounds are included in the topic Library of compounds, listed as: ribavirin (Sigma-Aldrich) was diluted to 20 mM in 100% dimethyl sulfoxide (DMSO; Sigma-Aldrich). Interferon α-2A (IFN-α 2A; Blau Farmacêutica, Cotia, São Paulo, Brazil) was diluted to 1000 IU/ml. (line 454-457)
- The procedure for compound dilution and treatment consistent across tests? Are the flow cytometry and apoptosis assays described in sufficient detail?
- All these methodologies are described in the supplementary material of this manuscript.
- The method for calculating CC50, IC50, and SI values should be adequately explained in the MS.
- The explanations were included in the methodology topic entitled "Screening for the anti-MAYV activity of the compounds".
- The statistical analysis should be properly explained appropriately for the type of data collected.
- We included the following sentence in the text: “Statistical evaluation was performed utilizing analysis of variance (ANOVA) followed by Tukey's test, with a significance level set at p<0.05. Nonlinear regression analysis, followed by the variable slope model, was employed to calculate CC50 and IC50 values”.
- The virucidal, adsorption, and internalization assays should clearly be defined and justified.
- Virucidal, adsorption and internalization tests are described in details in the supplementary methodology.
- The procedures for testing multiple MAYV genotypes sound should be discuss properly.
- We thank for your suggestion. The following sentence was added to the main text “Previous phylogenetic analysis revealed three MAYV genotypes: ‘D,’ ‘L,’ and ‘N’. To assess whether the effects of the selected compounds are genotype-specific, we first established infection parameters by analyzing the percentage of infected cells with two MAYV strains: genotype ‘D’ (MAYV_D; Figure 6B; blue bars) and genotype ‘L’ (BeAr20290; Figure 6B; bars red). Infection with the MAYV BeAr20290 strain in Huh7.5 cells was conducted using an MOI of 0.1 over 24 hpi. (line 396-413).
- the findings on antiviral activity statistically are significant and well-supported. Do the selected compounds show consistent effects across different tests? Please comaptre your results with reported one
- Thank you for the comment. When compared with tests such as flow cytometry, we have similar results with the reported ones. However, it is worth mentioning that the methodology used in our study is more sensitive than flow cytometry. In addition, the assays were standardized using both approaches and the results were consistent, although, as mentioned, the iHTS assay was more sensitive.
- the absence of virucidal activity should be clearly interpreted and explained
please see above.
- Does the differences in compound effectiveness across time-points clearly visualized?
- The differences in compound effectiveness are demonstrated in the graphs according to their addition time. There are clear differences in all treatments, both in data values, as visually, in microscopy images.
- Does the results across different human cell lines support the main conclusions?
- Yes. There is a continuous effect regardless of the cell line used and independent of the viral genotype used.
- Does the discussion adequately interpret the results in light of the hypothesis?
- Yes, it supports our hypothesis.
- Are the antiviral mechanisms of action discussed in a meaningful way? Is the comparison with previous studies thorough and accurate?
- Yes, the comparison with previous studies is complete and accurate, as well as comparing the effects of the compounds in other related arboviruses.
- Are the roles of flavonoids and carbamates discussed in terms of structure-activity relationship? Are alternative explanations for the data considered?
- Thank you for this consideration. Since carbamates did not show promising results, we did not discuss them in comparison with flavonoids. However, the method of obtaining the compounds was added to the supplementary material.
- At page-25, there are three -OH group, How you can protect the remaining two -OH group of naringenin and chrysin derivatives.
- The 5-OH hydroxyl groups of flavonoids present low acidity because of a strong intramolecular hydrogen bonding with the carbonyl group at C4 through a six-membered ring. Thus, drastic reaction conditions and higher temperatures are necessary for to achieve 5O-functionalization, which does not occur under our reactional conditions.
- Did you characterize the synthesized compound through HNMR, CNMR, HRMS, and FTIR. Please provide their spectral data of all synthesized compounds.
- The characterization data for the Naringenin derivatives (Scheme 1) and carbamates (Scheme 3) have already been reported in the literature and are cited throughout the text. Additionally, we have included the spectroscopic data for the chrysin derivatives (Scheme 2) in the SI.
Round 2
Reviewer 2 Report
Comments and Suggestions for Authors
Accept in present form.